# Genome-Wide Identification, Characterization, and Expression Profiling of TaDUF668 Gene Family in *Triticum aestivum*

Xiaohui Yin [1], Yi Yuan [1], Xiaowen Han [1], Shuo Han [1], Yiting Li [1], Dongfang Ma [1], Zhengwu Fang [1], Shuangjun Gong [2,3] and Junliang Yin [1,2,3,*]

1 MARA Key Laboratory of Sustainable Crop Production in the Middle Reaches of the Yangtze River (Co-Construction by Ministry and Province), College of Agriculture, Yangtze University, Jingzhou 434025, China

2 Key Laboratory of Integrated Pest Management of Crops in Central China, Ministry of Agriculture, Institute of Plant Protection and Soil Science, Hubei Academy of Agricultural Sciences, Wuhan 430064, China

3 Hubei Key Laboratory of Crop Diseases, Insect Pests and Weeds Control, Institute of Plant Protection and Soil Science, Hubei Academy of Agricultural Sciences, Wuhan 430064, China

* Correspondence: yinjunliang@yangtzeu.edu.cn

**Abstract:** *DUF668s*, a plant-specific gene family, encode proteins containing domain of unknown function (DUF) domains. Despite their essential functions, there is a lack of insight into *Triticum aestivum* TaDUF668s. Here, 31 *TaDUF668s* were identified from the wheat genome; according to phylogenetic relationships, they were named *TaDUF668-01* to *TaDUF668-31*. All TaDUF668s were hydrophilic and unstable proteins. There were 22 TaDUF668s that showed subcellular localization in nucleus. Evolutionary analysis demonstrated that TaDUF668s had undergone strong purifying selection, and fragment duplication plays major role in TaDUF668 family expansion. *Cis*-element prediction displayed that over 90% of TaDUF668 promoter regions contain the growth and abiotic responsiveness element. Consistently, expression profiling showed that *TaDUF668s* were highly induced in five wheat growth and development stages, seven main different tissues, five abiotic stresses, and five pathogenic stresses. In total, 12 *TaDUF668s* were targeted by 20 miRNAs through the inhibition of translation and cleavage patterns. RT-qPCR results confirmed that the expression of six *TaDUF668s* was significantly regulated by NaCl, PEG, *F. graminearum*, and *P. striiformis*; nevertheless, the regulation patterns were different. In summary, through systematic identification, characterization, evolutionary analysis, and expression profiling, a comprehensive understanding of *TaDUF668* has been obtained, which lays a foundation for further functional studies of *TaDUF668*.

**Keywords:** bioinformatics; *cis*-element; expression profiling; gene structure; RT-qPCR; subcellular localization; wheat

## 1. Introduction

Wheat (*Triticum aestivum* L.) is widely accepted as one of the most important food crops all over the world [1]. With the rapid growth of population in recent years, food production must keep up with population growth, especially for main food crops such as wheat [2]. However, the field production of wheat is seriously threatened by a variety of biotic and abiotic stresses, such as bacteria, fungi, nematodes, drought, salt, and extreme temperature, which heavily impact the production and quality of wheat [3]. Cultivating varieties with a tolerance for stresses is the most economical and fundamental way to decrease losses [4], and discovery of stress resistance genes is the fundamental for resistance breeding.

The domain of unknown function (DUF) is an unknown function protein domain in the protein family database [5]. The Pfam database (version 35.0) currently includes 19,632 gene families, in which almost 24% (4795) consist of DUF families [6]. In recent years, the rapid development of genomics and proteomics has provided important information

and powerful tools for systematic identification and characterization of DUF family proteins, which further set the basis for deciphering the biological roles of DUF proteins in regulating plant growth and development, as well as responding to biotic and abiotic stresses [7,8]. For instance, in Arabidopsis, DUF506 family members respond to environmental changes and participate in the calcium-signaling pathway [9]. In wheat, *TaWTF1* encodes a protein containing the DUF860 domain; the promoter region of this gene contains stress and hormone response elements, and its expression is upregulated by high-temperature stress in seedling and flowering stages [10]; *TaDUF966-9B* knockout showed severe leaf rolling symptoms under salt stress, indicating that *TaDUF966-9B* played a positive regulatory role in the response process of wheat to salt stress [7]. *TaCRR1* encodes a protein consisting of two DUF26 domains, and its expression was upregulated in wheat after infection by *Bipolaris sorokiniana*; meanwhile, its exogenous expression products can inhibit the growth of *B. sorokiniana* [11]. The protein encoded by *TaGW5*, a gene of the DUF4005 family, is involved in the young spike development [12].

DUF668 is a protein family that is widely found in monocots, dicots, mosses, and other species [13]. Its family members contain a conserved domain composed of 29 amino acids. Previous studies have shown that DUF668 is not only closely related to plant growth and development but also plays a vital regulatory effect in adverse situation response [13]. For example, in *Oryza sativa*, *OsDUF668-3* is highly expressed in leaves, roots, immature seeds, glumes, and panicles at 20 days after flowering, and *OsDUF668-1, -3, -4,* and *-5* are upregulated when rice is subjected to mechanical damage and infected by *Pirospora griseum* [13]. In *Gossypium hirsutum*, *GhDUF668-05, -08, -11, -23,* and *-28* were significantly induced under drought and verticillium wilt conditions and participated in the process of stress response [8]. In *Ipomoea batatas*, *IbDUF668-6, -7, -11,* and *-13* were upregulated in response to ABA, drought, and salt stress [14].

DUF668 family members have been identified and analyzed in *O. sativa* [13], *G. hirsutum* [8], and *I. batatas* [14], and their stress-responsive patterns have been studied. However, systematic understanding of wheat TaDUF688 family members is lacking to date. Therefore, in this study, we systematically identified TaDUF668 family members, characterized protein features, and analyzed their expression patterns under several stresses, with the aim of providing insights into *TaDUF668* and laying a theoretical basis for further deciphering their biological function.

## 2. Materials and Methods

### 2.1. Identification and Phylogenetic Analysis of TaDUF668

The reference genome and protein sequences of common wheat were collected from the International Wheat Genome Sequencing Consortium website (https://wheat-urgi.versailles.inra.fr/Seq-Repository/Assemblies/ accessed on 15 August 2023) [15]. The hidden Markov model (HMM) of DUF668 (PF05003) was downloaded from Pfam (http://pfam.xfam.org/ accessed on 15 August 2023) [16], and 6 DUF668 protein sequences from *Arabidopsis thaliana* [13], 12 sequences from *O. sativa* [13], 14 sequences from *Zea mays* [13], and 32 sequences from *G. hirsutum* [8] were collected. They were merged and used as reference sequences to perform BLASTp (e-value $< 1 \times 10^{-5}$) searching against the wheat protein sequences to identify the candidate TaDUF668s. After checking with Pfam (v35.0, http://Pfam.xfam.org/ accessed on 15 August 2023) and InterProScan (v94.0, http://www.ebi.ac.uk/InterProScan/ accessed on 15 August 2023), candidates containing DUF668 domain were extracted. After eliminating redundant sequences and incompletely annotated sequences, the remaining sequences were considered as TaDUF668 family members [8].

Multiple sequence alignment of TaDUF668 proteins was performed by neighbor-joining (1000 replicated bootstraps) using ClustalW2 [17]. A phylogenetic tree containing TaDUF668s, AtDUF668s, OsDUF668s, ZmDUF668s, and GhDUF668s was drawn using the iTOL (Interactive Tree of Life, http://itol.embl.de/ accessed on 15 August 2023) [3].

TaDUF668 members were classified and named according to their phylogenetic relationships [3].

### 2.2. Chromosome Localization and Interspecific Evolutionary Analysis of TaDUF668

The position information of TaDUF668s was obtained from the GFF3 file, and the chromosome distribution map was drawn using MapInspect [18]. To investigate the type of gene replication events, TaDUF668 collinearity analysis was performed using TBtools [19]. DUF668s of *Aegilops tauschii*, *Triticum dicoccoides*, and *Triticum urartu* were identified using the same strategy as wheat TaDUF668s. DUF668s of four species were merged and analyzed by BLASTn to find orthologous gene pairs (e-value $< 10^{-5}$, similarity $> 80\%$) [20]. After removing duplicate pairs, TBtools was used to calculate the Ka (nonsynonymous substitution rate), Ks (synonymous substitution rate), and Ka/Ks values [20] between *DUF668* gene pairs of four species.

### 2.3. TaDUF668 Gene Structure and Conserved Motif Analysis

According to the GFF3 gene structure annotation information, TBtools was used to draw the exon/intron structure map of *TaDUF668* genes [21]. The conserved motif of TaDUF668s was identified using the online website MEME (version 5.5.2, http://meme-suite.org/tools/meme/ accessed on 15 August 2023) [22]. The parameters were set as follows: each protein sequence can incorporate any number of non-overlapping protein motifs, the quantity of different protein motifs is 10, and the width of protein motifs ranges from 6 to 50 amino acids. TBtools was used to analyze the output results and draw the protein motif structure map [21].

### 2.4. Characteristics, Three-Dimensional Structure Prediction, and Subcellular Localization of TaDUF668 Proteins

Using ExPASySenver10 (https://prosite.expasy.org/PS50011/ accessed on 15 August 2023), the physical and chemical properties of TaDUF668s, including protein sequence length (Len), the molecular weight (MW), the isoelectric point (pI), and total hydrophilicity (GRAVY) were analyzed [23]. Through the Plant-mPLoc (http://www.csbio.sjtu.edu.cn/bioinf/Plant/ accessed on 15 August 2023), subcellular localization of TaDUF668s was predicted [24]. A three-dimensional structure of TaDUF668s was built using SWISS-MODEL (https://www.swissmodel.expasy.org/ accessed on 15 August 2023) [25].

### 2.5. Cis-Regulatory Element Analysis of TaDUF668

To analyze the *cis*-acting elements in the promoter region of the *TaDUF668* genes, the upstream promoter sequence (1–1500 bp) of *TaDUF668s* was manually extracted from the reference genome. Then, sequences were uploaded into PlantCARE website (http://bioinformatics.psb.ugent.be/webtools/plantcare/html/ accessed on 15 August 2023) to identify *cis*-elements. The analysis results were organized and presented using the R package "pheatmap" [26].

### 2.6. Expression Profiling of TaDUF668s

Wheat transcriptome sequencing data were downloaded from the NCBI-SRA database and compared with the wheat reference genome using Hisat2 [7]. The expression level of *TaDUF668* genes (represented by normalized TPM value) was calculated by Cufflinks. The $\text{Log}_2(\text{TPM} + 1)$ value was used to plot a heatmap through the R package "pheatmap" to exhibit the expression profiling of *TaDUF668* genes under different conditions [7].

### 2.7. Prediction of the Targeting Relationship between miRNAs and TaDUF668

To identify miRNAs targeting *TaDUF668s* transcripts, mature sequences of wheat miRNAs (http://www.mirbase.org/ accessed on 15 August 2023) were collected from a previous study [7]. Then, the miRNA and TaDUF668s CDS sequences were submitted to psRNA Target (https://www.zhaolab.org/psRNA/ accessed on 15 August 2023) [15,27]

to analyze the targeting relationships between TaDUF668s and miRNAs. The R packages "ggplot2" and "ggalluvial" were used to draw the targeting relationship map between miRNA and *TaDUF668s* [28].

### 2.8. Stress Treatment of Wheat

Wheat seeds of cultivar Yangmai 20 were surface-sterilized with 1% sodium hypochlorite solution then washed completely using double-distilled water and cultured for two days at 25 °C in Petri dishes, which were lined with two layers of saturated filter paper [29]. The germinated seeds were then transplanted into a 25% concentration Hoagland solution. After three days, we increased the density of solution to 50% and set growth conditions to a 16/8 h (day/night) photoperiod at 25 °C [1]. According to the methods reported by Yang et al. [30], *Fusarium graminis* (PH-1) spores were obtained. When the seedlings reached one leaf stage, 10 μL of the spore suspension ($5 \times 10^5$ cells/mL) was drawn with a pipette and dropped onto the leaves. They were then wrapped in wet paper towels to retain moisture and incubated at 25 °C at 65% relative humidity. Wheat leaves were harvested at 6 h, 12 h, 24 h, 48 h, 72 h, and 96 h after inoculation, and uninoculated wheat seedlings were used as controls. Referring to the methods of Zhan et al. [31], the spore suspension of *Puccinia striiformis* were gained. Afterwards, we inoculated wheat leaves with stripe rust spore suspension. Leaves were taken at 6 h, 12 h, 24 h, and 48 h after inoculation and using uninoculated wheat as a control [31]. In the meantime, 150 mM NaCl and 20% PEG6000 solution were used to treat wheat, and double-distilled water was used as a control. Growth conditions were a 16 h/8 h (day/night) photoperiod at 25 °C. Leaves were collected at 2 h, 6 h, 12 h, 24 h, 48 h, and 72 h after treatment [1,32]. These experimental samples were saved in liquid nitrogen right away and preserved at −80 °C until use. Each sample contained at least three biological replications.

### 2.9. Real-Time Quantitative PCR

Total RNA was extracted from each wheat leaf sample using TRIzol reagent (GenStar, Beijing, China). RNA was reverse-transcribed into cDNA using HiScript II 1st Strand cDNA Synthesis Kit (+gDNA wiper) (Vazyme, Nanjing, China). Six *TaDUF668* genes, which respond to multiple stresses, were selected for the RT-qPCR analysis. The primers were designed using Primer Premier 5.0 (Table 1). The RT-qPCR was initiated on a CFX 96 Real-Time PCR system (Bio Rad, Hercules, CA, USA) using ChamQ SYBR qPCR Master Mix (Vazyme, Nanjing, China). *Ta2291* was used as the internal reference gene for RT-qPCR analysis, which is ADP-ribosylation factor, and its expression level was stable with different treatments [1]. Three independent biological replicates and three independent technical replicates were performed for each treatment and control sample. Gene expression was calculated using a $2^{-\Delta\Delta Ct}$ method [7].

**Table 1.** RT-qPCR primers for six *TaDUF668* genes.

| Gene Name | Forward Primer | Reverse Primer |
|---|---|---|
| *TaDUF668-09* | GGGAACCACCCGAGGAAATA | TATCCGCACGCCATCTGAAT |
| *TaDUF668-11* | GCAGGAAGTGAAGAGCCAAAGT | ATGACCAAAGGCGTCGTAAATC |
| *TaDUF668-14* | TTCTCGCTCATCTTCTGTTCCT | CAAGCCACCGTAATGTCTTCTC |
| *TaDUF668-26* | GCGGAAGCAACGGTTCAG | CAACGAGCACCTGGACGAT |
| *TaDUF668-27* | GGACGAGCCAAGCGAGAAG | CTGTCAACGGAGGTGGCAAT |
| *TaDUF668-28* | GGACGAGCCAAGCGAGAAG | CTGTCAACGGAGGTGGCAAT |

In order to investigate the normality between RT-qPCR gene expression data, skewness and kurtosis tests were used to determine the normality between the data through a data processing system (DPS). One-factor analysis of variance was used to determine significant differences between the means through the DPS (Tang Qi-Yi, Zhejiang University, Hangzhou, China).

## 3. Results

### 3.1. Identification and Phylogenetic Analysis of TaDUF668s

Through a systematic bioinformatics analysis, 31 *TaDUF668* genes were identified from *T. aestivum*. To determine the relationships between *A. thaliana*, *O. sativa*, *Zea mays*, and *G. hirsutum* DUF668s, a phylogenetic tree containing 95 DUF668 protein sequences was constructed. As demonstrated in Figure 1, TaDUF668s showed the closest evolutionary relationships with OsDUF668s. Thus, following the grouping method of OsDUF668s, 31 TaDUF668s were divided into two groups. Group I contained 15 TaDUF668 members, and Group II contained 16 TaDUF668 members; according to the evolutionary relationships shown in the phylogenetic tree, they were named TaDUF668-01 to TaDUF668-31.

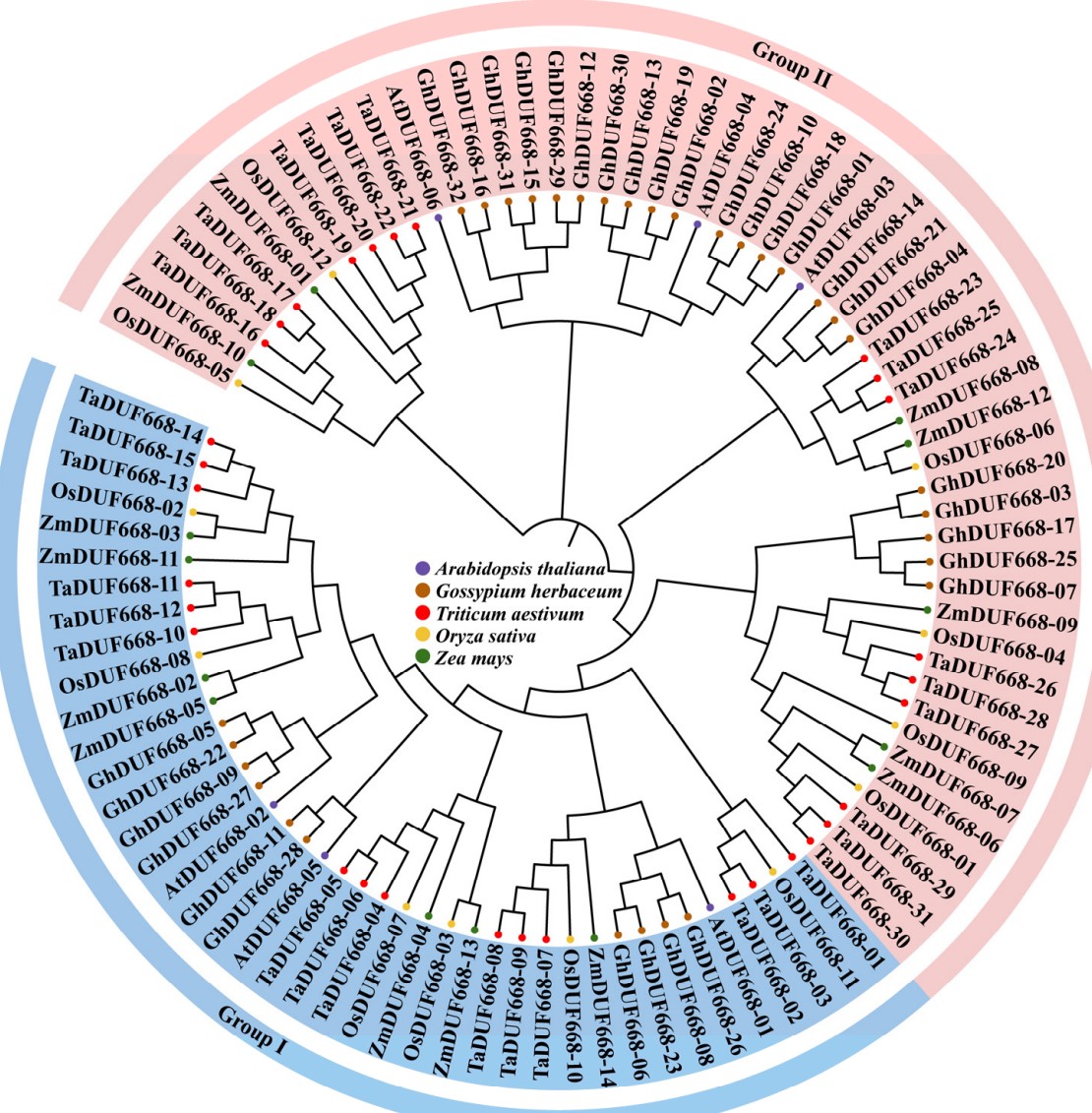

**Figure 1.** Phylogenetic tree of *Triticum aestivum*, *Arabidopsis thaliana*, *Gossypium hirsutum*, *Oryza sativa*, and *Zea mays* DUF668s.

### 3.2. Chromosome Localization and Evolution Analysis of TaDUF668s

As shown in Figure 2A, 31 *TaDUF668* genes were distributed on 18 chromosomes and were not uniformly distributed. Chr1A, Chr2A, Chr7A, Chr1B, Chr2B, Chr7B, Chr1D, Chr2D, Chr4D, and Chr7D had the fewest *TaDUF668s* with only one, while Chr5B had the

most *TaDUF668s* with five. Among them, there were 11, 11, and 9 *TaDUF668* genes in sub genomes A, B, and C, respectively.

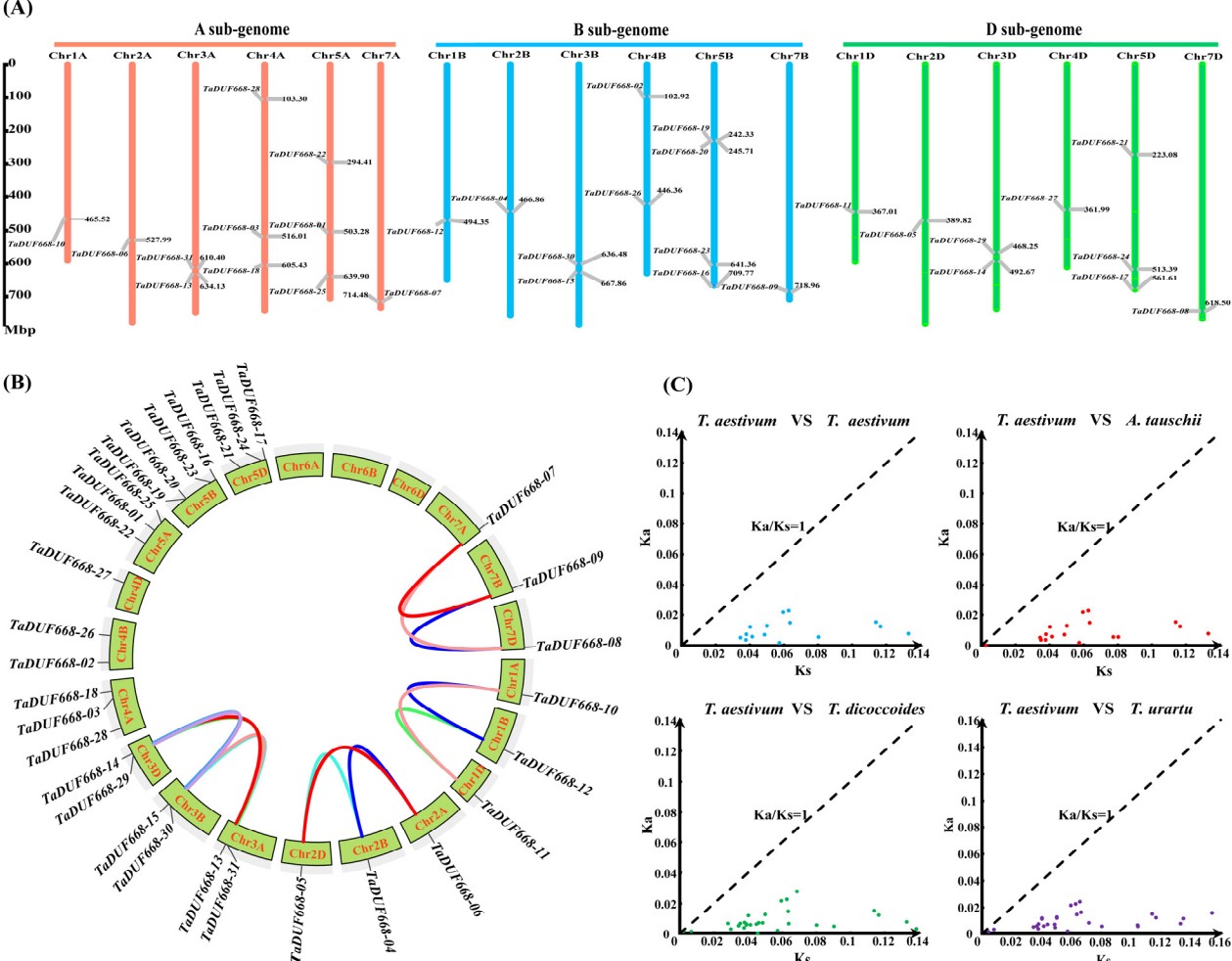

**Figure 2.** Chromosomal localization and evolution analysis of TaDUF668s. (**A**) Distribution of chromosomal localization of *TaDUF668* genes. The scale mark on the left denotes the physical map distance among genes (Mbp). Chromosomes are indicated by thick vertical lines of different colors and lengths. (**B**) collinearity analysis for *TaDUF668* genes in *T. aestivum*. (**C**) Ka and Ks scatter diagram of DUF668 homolog gene pairs among *T. aestivum* and three ancestors of wheat.

According to the sequence similarity and chromosome location, as depicted in Figure 2B, a total of 15 pairs of fragment duplications were discovered among 31 *TaDUF668* genes, and no tandem duplication event was discovered. The result indicated that fragment duplication plays a major role in the expansion of the TaDUF668 family.

In order to deeper understand the evolutionary history of *TaDUF668* genes, we worked out the Ka/Ks rates of 15 pairs of TaDUF668 fragment duplications using TBtools. Meanwhile, 15 (*T. aestivum* vs. *T. aestivum*), 18 (*T. aestivum* vs. *A. tauschii*), 33 (*T. aestivum* vs. *T. dicoccoides*), and 30 (*T. aestivum* vs. *T. urartu*) homologous gene pairs were identified. The Ka/Ks ratio was calculated by TBtools. As depicted in Figure 2C, the Ka/Ks value between all homologous gene pairs was less than 1, which indicated that *DUF668s* had undergone strong purifying selection pressure among common wheat and the three ancestral wheat species.

### 3.3. TaDUF668 Gene Structure and Conserved Motifs

Based on the annotation information in GFF3 file, the *TaDUF668* exon/intron structure was analyzed using TBtools. As exhibited in Figure 3B, the CDS regions of *TaDUF668* genes ranged from 1 to 13. Interestingly, among the members in Group II, only *TaDUF668-27*, and *-28* contained one intron, whereas all Group I members had multiple introns. In addition, *TaDUF668-01*, *-05*, *-06*, *-07*, *-08*, *-09*, *-10*, *-13*, *18*, *-20*, *-21*, *-22*, *-26*, *-27*, *-28*, and *-31* contained UTR regions at both the 5′ and 3′, and *TaDUF668-04* only contained UTR regions at the 3′. Conserved motif analysis showed that TaDUF668s contain conserved motifs ranging from 3 to 10 (Figure 3B). TaDUF668-01 contains the fewest motifs (3), and TaDUF668-04 to TaDUF668-15 contains the most motifs (10). Among them, Motifs 6, 7, and 8 constituted the typical DUF668 domain; Motifs 1 and 3 constituted the DUF3475 domain; the remaining motifs did not match the known functional domains. In addition, combined with phylogenetic analysis, it was found that members in the same branch had similar gene structure and conservative motifs.

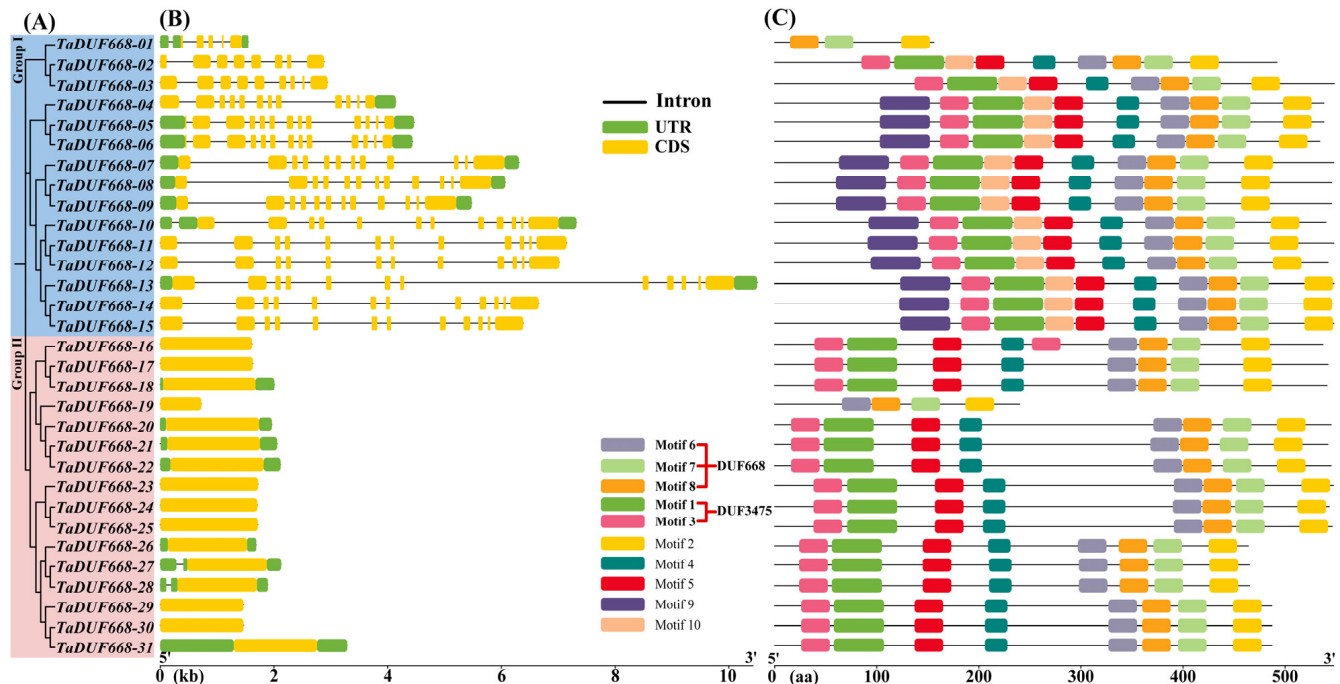

**Figure 3.** TaDUF668 gene structure and conserved protein motif diagram. (**A**) TaDUF668 phylogenetic tree. (**B**) *TaDUF668* gene structure. Yellow and green squares represent the CDS sequence and UTR non-coding regions, respectively, while black lines represent introns. (**C**) TaDUF668 protein motif distribution.

### 3.4. TaDUF668 Protein Characteristics, Subcellular Localization, and 3D Structure Prediction

As exhibited in Table 2, the length of the TaDUF668s ranged from the 156 amino acid (aa) (TaDUF668-01) to 667 aa (TaDUF668-13) length, with an average of 537.87 aa. The protein molecular weight ranged from 17.45 (TaDUF668-01) to 74.07 (TaDUF668-13) kDa, with a mean value of 59.38 kDa. The isoelectric points ranged from 6.43 (TaDUF668-01) to 9.96 (TaDUF668-26), with an average of 8.77. All proteins were basic (pI > 7) except TaDUF668-01. The protein instability index ranged from 39.64 (TaDUF668-19) to 62.02 (TaDUF668-02), with an average value of 51.99. All TaDUF668s were unstable proteins (instability index > 40) except TaDUF668-19. The hydrophilic coefficients of TaDUF668s ranged from −0.707 (TaDUF668-08) to −0.096 (TaDUF668-01), all of which were less than zero, indicating that TaDUF668s were hydrophilic. No signal peptide was detected, indicating that all TaDUF668s were non-secreted proteins. Subcellular localization prediction demonstrated that 29% of TaDUF668 proteins were predicted to be localized only in chloro-

plasts; 58% of TaDUF668s were predicted to be localized only in the nucleus. While 10% of the TaDUF668s were localized in the chloroplast, nucleus, and cell membrane, TaDUF668-03 was predicted to be simultaneously localized in the nucleus, cytoplasm, and chloroplast.

**Table 2.** Protein characteristics of TaDUF668s.

| Name | Gene ID | Len. | MW. | pI. | Ins. | GRAVY | Sig. | Sub. |
|---|---|---|---|---|---|---|---|---|
| TaDUF668-01 | TraesCS5A03G0714700 | 156 | 17.45 | 6.43 | 44.15 | −0.10 | No | Nuc |
| TaDUF668-02 | TraesCS4B03G0220500 | 492 | 56.11 | 9.48 | 62.02 | −0.59 | No | Nuc |
| TaDUF668-03 | TraesCS4A03G0577300 | 552 | 62.15 | 8.87 | 55.17 | −0.56 | No | Chl/Nuc/Cyt |
| TaDUF668-04 | TraesCS2B03G0837000 | 538 | 59.50 | 8.66 | 43.69 | −0.33 | No | Chl |
| TaDUF668-05 | TraesCS2D03G0700600 | 538 | 59.74 | 8.73 | 44.08 | −0.37 | No | Chl |
| TaDUF668-06 | TraesCS2A03G0759400 | 534 | 59.33 | 8.80 | 41.33 | −0.32 | No | Chl |
| TaDUF668-07 | TraesCS7A03G1285000 | 624 | 70.07 | 8.98 | 57.30 | −0.70 | No | Nuc |
| TaDUF668-08 | TraesCS7D03G1219600 | 622 | 69.85 | 8.97 | 59.03 | −0.70 | No | Nuc |
| TaDUF668-09 | TraesCS7B03G1202200 | 622 | 69.66 | 8.95 | 58.49 | −0.70 | No | Nuc |
| TaDUF668-10 | TraesCS1A03G0682400 | 645 | 72.40 | 7.18 | 52.38 | −0.52 | No | Nuc |
| TaDUF668-11 | TraesCS1D03G0646400 | 644 | 72.05 | 7.44 | 52.28 | −0.50 | No | Nuc |
| TaDUF668-12 | TraesCS1B03G0776500 | 647 | 72.53 | 7.74 | 52.66 | −0.52 | No | Nuc |
| TaDUF668-13 | TraesCS3A03G0907200 | 667 | 74.07 | 9.17 | 54.52 | −0.68 | No | Nuc |
| TaDUF668-14 | TraesCS3D03G0833600 | 666 | 73.96 | 9.16 | 55.08 | −0.68 | No | Nuc |
| TaDUF668-15 | TraesCS3B03G1028600 | 667 | 74.06 | 9.16 | 54.80 | −0.68 | No | Nuc |
| TaDUF668-16 | TraesCS5B03G1360500 | 537 | 57.58 | 9.61 | 48.32 | −0.18 | No | Nuc |
| TaDUF668-17 | TraesCS5D03G1204500 | 542 | 57.93 | 9.67 | 49.08 | −0.20 | No | Nuc |
| TaDUF668-18 | TraesCS4A03G0791000 | 541 | 57.95 | 9.67 | 48.51 | −0.21 | No | Nuc |
| TaDUF668-19 | TraesCS5B03G0355900 | 240 | 26.25 | 6.50 | 39.64 | −0.40 | No | Nuc |
| TaDUF668-20 | TraesCS5B03G0360900 | 545 | 59.58 | 9.65 | 47.45 | −0.25 | No | Chl |
| TaDUF668-21 | TraesCS5D03G0347700 | 542 | 59.32 | 9.62 | 47.93 | −0.22 | No | Chl |
| TaDUF668-22 | TraesCS5A03G0365500 | 545 | 59.49 | 9.53 | 47.11 | −0.23 | No | Chl |
| TaDUF668-23 | TraesCS5B03G1143800 | 573 | 62.51 | 7.03 | 60.36 | −0.27 | No | Nuc |
| TaDUF668-24 | TraesCS5D03G1034300 | 569 | 62.12 | 7.63 | 59.34 | −0.24 | No | Nuc |
| TaDUF668-25 | TraesCS5A03G1081600 | 571 | 62.29 | 7.30 | 58.04 | −0.23 | No | Nuc |
| TaDUF668-26 | TraesCS4B03G0579600 | 464 | 50.95 | 9.96 | 55.67 | −0.33 | No | Chl |
| TaDUF668-27 | TraesCS4D03G0514500 | 465 | 51.17 | 9.84 | 57.11 | −0.35 | No | Chl |
| TaDUF668-28 | TraesCS4A03G0195500 | 465 | 51.19 | 9.92 | 55.40 | −0.35 | No | Chl |
| TaDUF668-29 | TraesCS3D03G0788700 | 487 | 53.13 | 9.37 | 50.59 | −0.22 | No | Cem/Chl/Nuc |
| TaDUF668-30 | TraesCS3B03G0984500 | 487 | 53.13 | 9.37 | 49.99 | −0.20 | No | Cem/Chl/Nuc |
| TaDUF668-31 | TraesCS3A03G0859800 | 487 | 53.13 | 9.37 | 50.04 | −0.21 | No | Cem/Chl/Nuc |

Len, length of amino acid (aa); MW, molecular weight (kDa); pI, isoelectric point; Ins, instability index; GRAVY, grand average of hydropathicity; Sig, signal peptide; Sub, subcellular localization; Nuc, nucleus; Chl, chloroplast; Cyt, cytoplasm; Cem, cell membrane.

SWISS-MODEL was used for 3D homology modeling of TaDUF668s. As shown in Figure 4, the 3D structures of 31 TaDUF668s are mainly composed of *α*-helices, *β*-turns, extended chains, and random coils. Among them, *α*-helices accounted for the largest proportion (35.42–61.94%), followed by random coils (26.08–40.42%) and extended strands (5.81–16.10%), and *β*-angles accounted for the smallest proportion (3.66–8.33%). The figure indicates that members within the same group have more similarities in 3D structure than members between different groups.

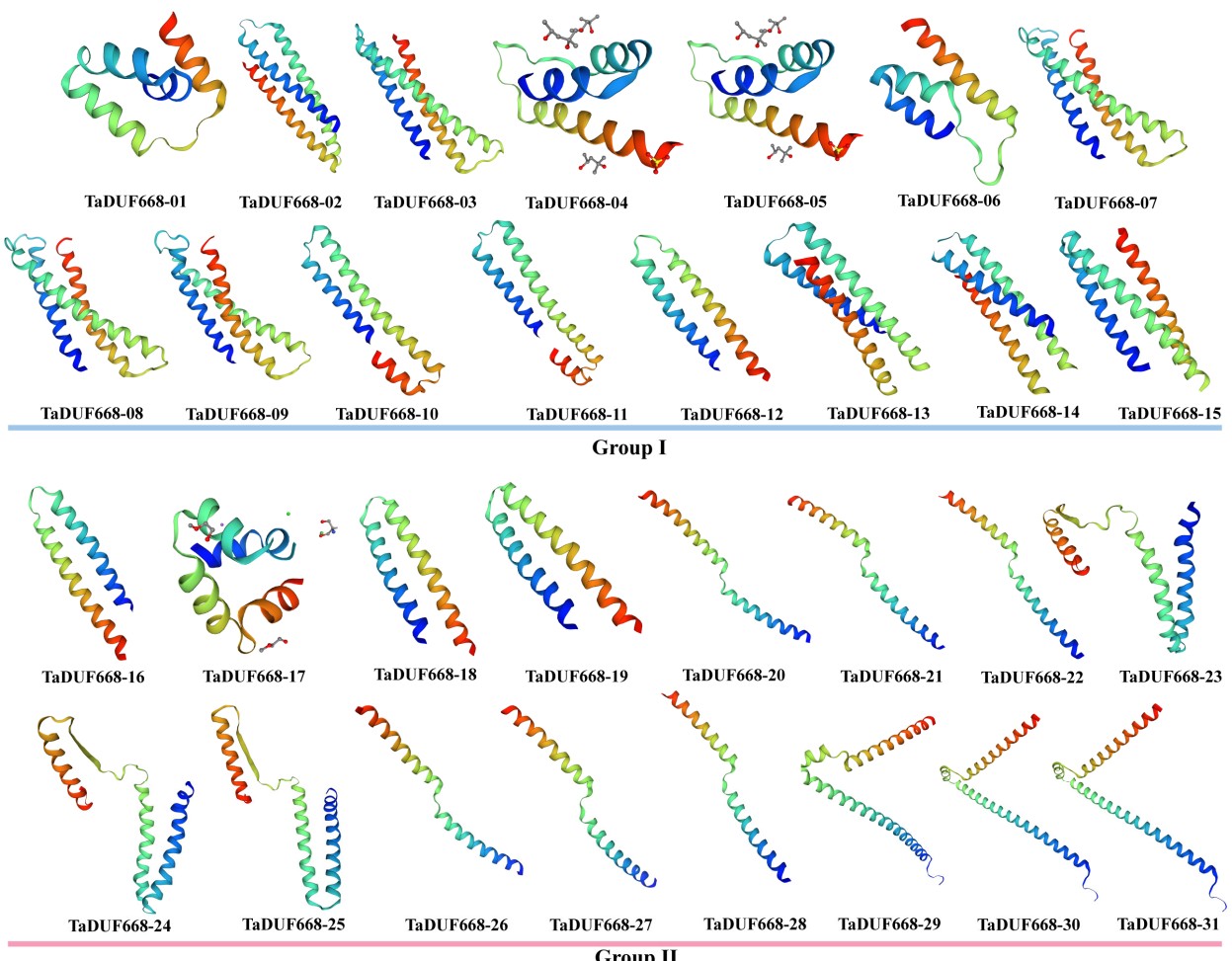

**Figure 4.** Predicted 3D models of TaDUF668s. Models were generated by using SWISS-MODEL.

*3.5. TaDUF668 cis-Element Analysis*

In this study, the upstream 1.5 kb regions of 31 *TaDUF668s* were analyzed and, in total, 50 kinds of *cis*-acting elements were identified. The results are depicted in Figure 5. They were divided into four categories: light response (20), abiotic stress (8), growth and development (9), and phytohormones (13). Among them, light-responsive elements are the most common (Figure 5A), including ACE, ATC-motif, AE-box, Box 4, Box II, CAG-motif, chs-CMA1a, chs-CMA2b, GA-motif, GATA-motif, Gap-box, G-box, GTGGC-motif, GT1-motif, I-box, MRE, Sp1, TCCC-motif, TCT-motif, and ATCT-motif. As exhibited in Figure 5B, stress response elements, such as anaerobic induction regulatory elements (GC-motif and ARE), drought response elements (MBSs), defense and stress-related elements (TC-rich repeats), and low-temperature response (LTR) elements, were also found in the TaDUF668 promoter. Meanwhile, growth- and development-related elements were also detected, such as the *cis*-element related to meristem expression (CAT-box), the seed-specific regulatory element (RY-element), the palisade mesophyll cell differentiation element (HD-Zip1), and cell cycle regulatory elements (MSA-like). As shown in Figure 5D, phytohormone response elements, such as auxin (AuxRR-core and TGA-element), gibberellin (GARE-motif and P-box), abscisic acid (ABRE), salicylic acid (TCA-element), and jasmonic acid (TGACG-motif and CGTCA-motif), were also detected.

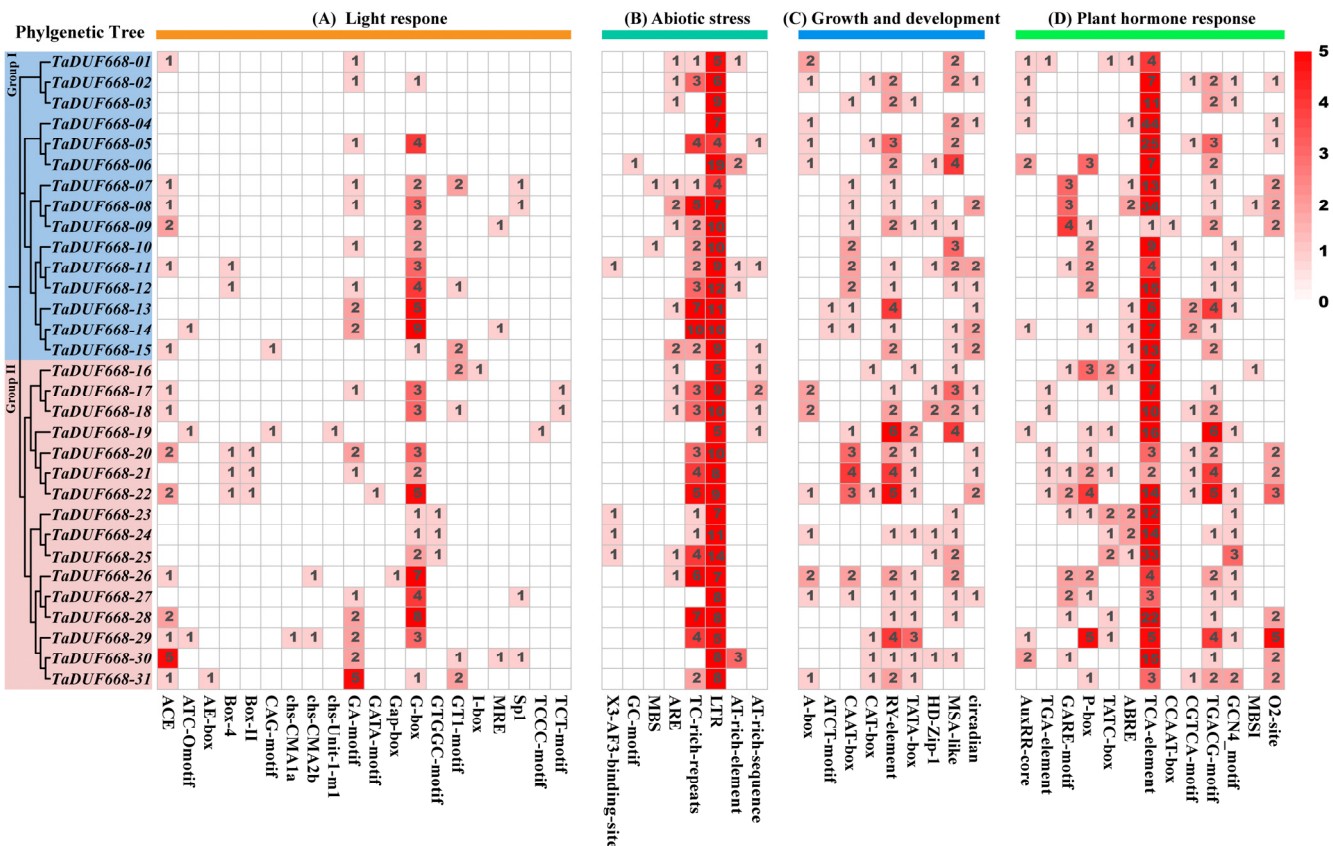

**Figure 5.** Distribution of the cis-acting elements detected in promoter regions of *TaDUF668*. The color and number of grids represent the number of cis-acting elements in the corresponding *TaDUF668*.

*3.6. TaDUF668 Expression Profiling Analysis*

In order to further explore the expression characteristics of *TaDUF668* genes, their expression patterns were analyzed using the RNA-seq transcriptomic data. As demonstrated in Figure 6A, the *TaDUF668* expression pattern was tissue-specific. The expression levels of *TaDUF668-10* to *TaDUF668-15* were generally higher throughout the growth and development stages, especially in root tissues. *TaDUF668-10, -11, -12, -13, -14,* and *-15* were highly expressed in roots, leaves, and spikes in Chinese Spring wheat. In contrast, the expression levels of *TaDUF668-10* to *TaDUF668-15* were lower in the Chinese spring wheat seeding stage and three leaf stage. *TaDUF668-11, -27,* and *-28* were highly expressed in flag leaves and leaf sheaths at the grain filling stage; the three genes were also highly expressed in roots of the three leaf stage and three leaf stage of Chinese spring wheat. But *TaDUF668-27* and *-28* were hardly expressed in grain, spikes, leaves, stems, and seedlings of Chinese spring wheat. As exhibited in Figure 6B, the expression levels of *TaDUF668-10* to *TaDUF668-15* in the root were higher than those in the shoot under phosphorus deficiency stress. After 1 h of drought treatment, the expression levels of *TaDUF668-14, -15,* and *-28* were higher than those in the control in the 7-day-old leaves of TAM107 wheat. The expression of *TaDUF668-11* was upregulated at 2 h and 12 h of drought treatment compared with the control. As depicted in Figure 6C, the expression levels of five *TaDUF668* (*TaDUF668-11, -14, -15, -27,* and *-28*) genes were highly induced in response to the stress of *Fusarium pseudograminearum*, *F. graminearum*, *Blumeria graminis*, *Puccinia striiformis*, and *Zymoseptoria tritici*. Among them, *TaDUF668-27* and *-28* were highly induced only by *B. graminis* and *P. striiformis* stresses. The expression levels of *TaDUF668-14* and *-15* were highly induced only by *Z. tritici* stress, whereas *TaDUF668-11* was highly induced only by *F. graminearum* stress. These results indicate that *TaDUF668* not only participates in wheat growth and development but also plays a momentous effect in wheat stress response.

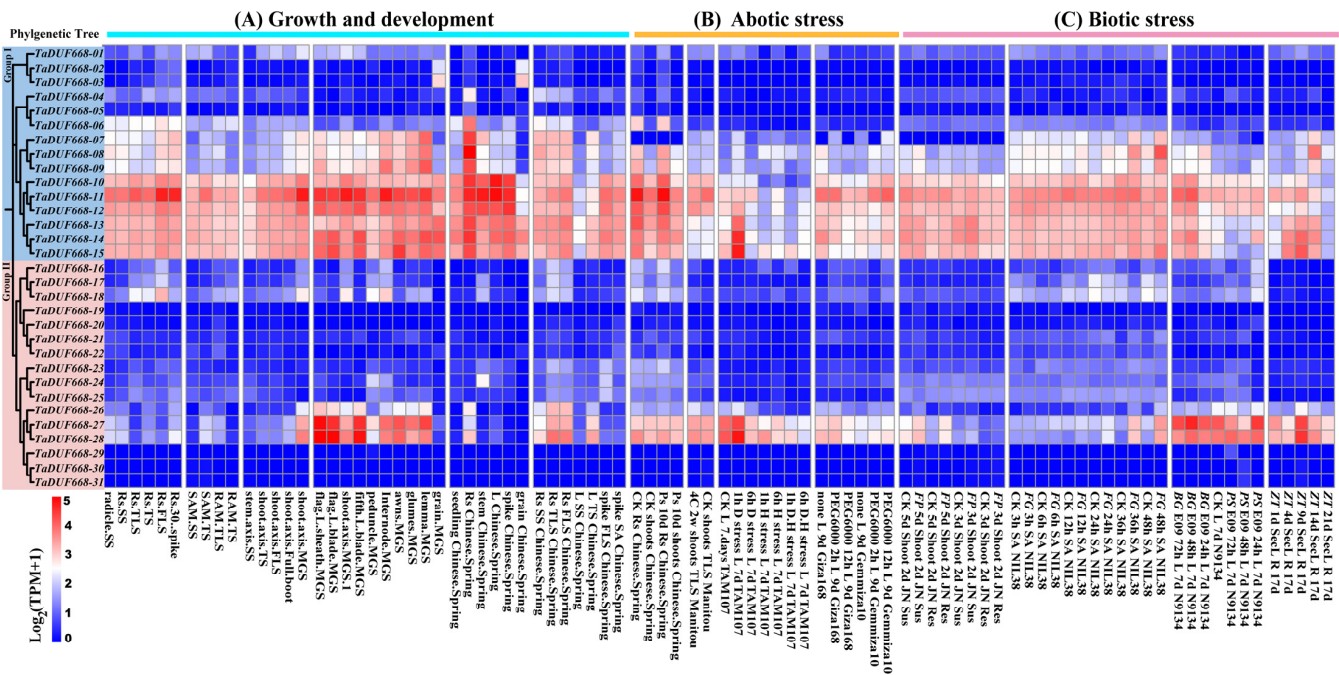

**Figure 6.** Expression patterns of *TaDUF668s* under growth and development as well as biotic and abiotic stress treatment conditions. Red indicates high expression level, and blue represents low expression level. Numbers from 0 to 5 indicate expression levels from low to high. L, leaf; Rs, roots; SAM, shoot apical meristem; RAM, root apical meristem; SS, seedling stage; TS, tillering stage; MGS, milk grain stage; TLS, three leaf stage; FLS, flag leaf stage; Ps, phosphorous starvation; D, drought; H, heat; D&H, drought and heat combined; Sus, susceptible; Res, resistant; *FG, Fusarium graminearum; FP, Fusarium pseudograminearum*; SA, spikelet anthesis; BG, *Blumeria graminis*; PS, *Puccinia striiformis*; ZT, *Zymoseptoria tritici*.

### 3.7. Post-Transcriptional Regulation of TaDUF668 by miRNAs

As can be seen from Figure 7, 20 wheat miRNAs were forecasted to target 12 *TaDUF668* gene transcripts through cleavage and translation repression. It is suggested that tae-miR6197-5p target *TaDUF668-11* and tae-miR9671-5p target *TaDUF668-14* by inhibiting translation; the remaining miRNAs target *TaDUF668s* by cleavage. Among them, tae-miR1119 and tae-miR9652-5p can simultaneously target *TaDUF668-13, -14,* and *-15*; tae-miR5086 targets *TaDUF668-18* and *-28*. Meanwhile, *TaDUF668-11* and *TaDUF668-18* were also targeted by tae-miR9677b and tae-miR9673 through cleavage. Interestingly, *TaDUF668-27* was targeted by four miRNAs, namely tae-miR10521, tae-miR1134, tae-miR9667-5p, and tae-miR9669-5p; *TaDUF668-28* was targeted by five miRNAs, namely tae-miR1124, tae-miR1136, tae-miR5086, tae-miR5384-3p, and tae-miR9666a-3p. Last but not least, *TaDUF668-06, -07, -08, -09,* and *-26* were targeted by tae-miR5085, tae-miR120b-3p, tae-miR1117, tae-miR9776, and tae-miR9676-5p, respectively. The results showed that multiple miRNAs degrade *TaDUF668* transcription products by cleavage or inhibit their translation process to form diverse regulatory relationships.

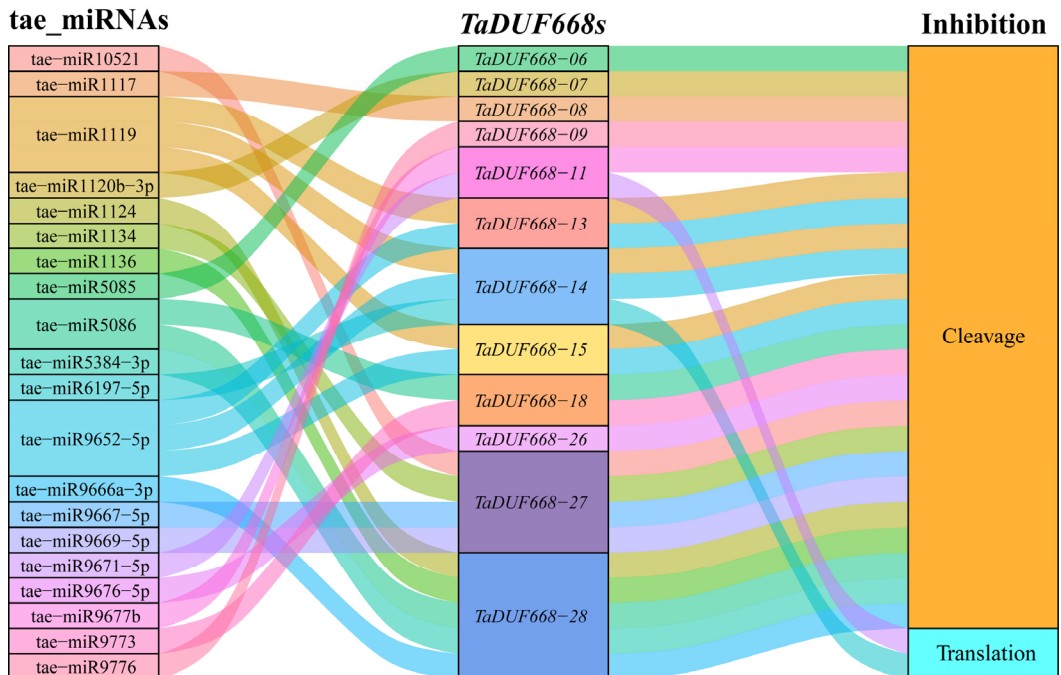

**Figure 7.** Sankey diagram for the relationships of miRNA targeting *TaDUF668* transcripts. The three columns represent miRNAs, *TaDUF668s*, and inhibition effect.

### 3.8. Real-Time Quantitative PCR Analysis

According to expression profiling results, *TaDUF668-09*, *-11*, *-14*, *-26*, *-27*, and *-28* were highly induced by *F. graminearum*, PEG6000, *B. graminis,* and *Z tritici* (Figure 8). To further validate the role of *TaDUF668* in wheat stress response, these six *TaDUF668* genes were selected for real-time PCR analysis. The expression levels of these six genes varied according to the treatment type and time.

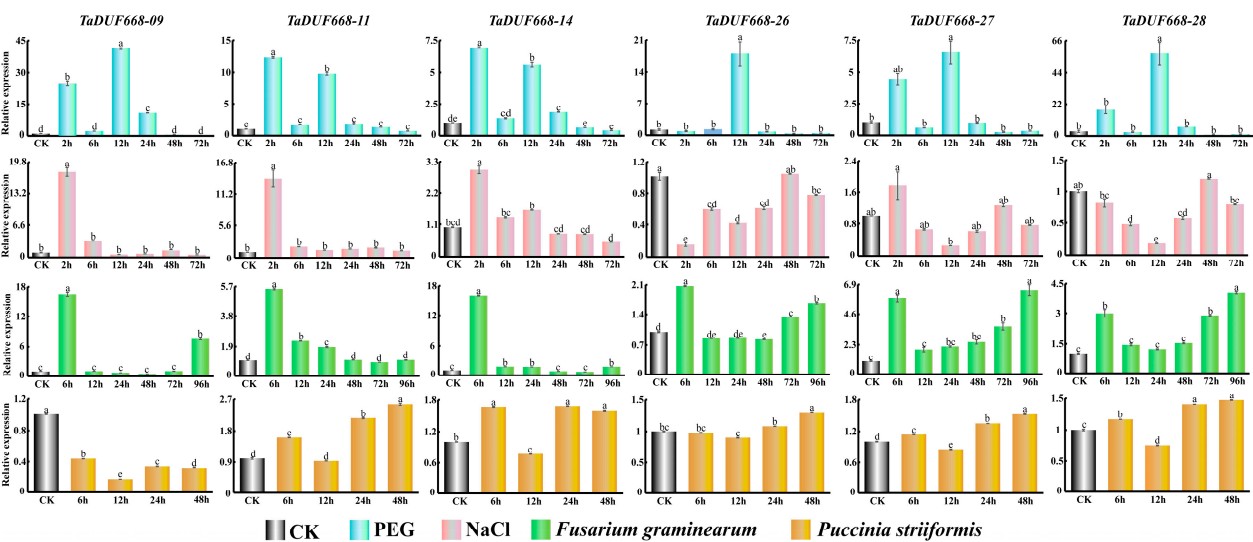

**Figure 8.** Expression patterns of six *TaDUF668* genes under different treatment conditions. The treatments include PEG, NaCl, *F. graminearum*, and *P. striiformis*. In the figure, the *Y*-axis represents the relevant expression level, and the *X*-axis represents the time point of stress treatment. The data from all experiments were expressed as the means ± SE. The results of tests were consistent with the approximate normal distribution. One-way analysis of variance was used to determine significant differences between the means. Error bars represents standard error. Different letters indicate means that are significantly different at the $p < 0.05$ level among different treatments conditions.

In PEG-treated leaves, *TaDUF668-09*, *-11*, *-14*, *-27*, and *-28* were highly upregulated at 2 and 12 h; among them, the expression level of *TaDUF668-09* was most significantly increased to forty-fold that of the CK control. Then, it returned to the CK level after 48 h. *TaDUF668-09* was only induced after 12 h and was consistent with the CK in rest time points.

In NaCl-treated leaves, four *TaDUF668* genes were highly upregulated at 2 h. Then, the expression level declined to the CK level after 6 h. However, the relative expression levels of *TaDUF668-26* and *TaDUF668-28* were similar to or lower than CK.

In leaves inoculated with *F. graminearum*, six *TaDUF668* genes were intensively upregulated at 6 h. The expression levels of *TaDUF668-09* and *TaDUF668-14* were higher than the other four *TaDUF668* genes. *TaDUF668-26*, *TaDUF668-27*, and *TaDUF668-28* were gradually ascended from 24 h to 96 h. Notably, the 96 h expression of *TaDUF668-27* was nearly six times higher than CK. On the contrary, *TaDUF668-11* and *TaDUF668-14* approximatively manifested a descending trend after 12 h. In particular, the expression level of *TaDUF668-09* was obviously elevated after 96 h.

In leaves inoculated with *P. striiformis*, except for *TaDUF668-09*, which had a lower expressed than CK at all time points, the expression levels of the genes (*TaDUF668-11*, *-14*, *-26*, *-27*, and *-28*) were lower than CK only at 12 h; thus, their expression was mostly highly induced after 6 h, 24 h, and 48 h. The increasing trend of *TaDUF668-11* and *-14* was clearly more pronounced than that *TaDUF668-26*, *-27*, and *-28*.

## 4. Discussion

Previous studies indicated that DUF668 plays important roles in plant growth and development and stress response in rice [13], cotton [8], and sweet potato [14]. Nevertheless, the DUF668 family in wheat, one of the most important main crops, is largely unknown. In order to provide an insight into wheat TaDUF668, we carried out systematic identification, characterization, and expression profiling analysis of family genes. In total, 31 *TaDUF668s* were identified (Figure 1 and Table 2). In comparison, there were 6, 11, 12, and 32 DUF668s in the reference genome of Arabidopsis [13], sweet potato [14], rice [13], and cotton [8]. Considering the ploidy and genome size, it seems that, during the evolution of wheat, the TaDUF668 family underwent the chromosome doubling process, and hexaploidy contributed to the family expansion. However, a few duplication genes may have been lost during evolution. The deduction explains the phenomenon for TaDUF668 whereby members are not evenly distributed in the chromosome (Figure 2A).

The Ka/Ks value, the ratio of non-synonymous replacement rate to synonymous replacement rate, is generally used to represent the rate of genetic evolution of a species [33]. A value equal to, larger than, or smaller than 1 represents a neutral, positive, or negative selection effect, respectively [34]. In this study, the result of evolution analysis manifested DUF668 family members of four species (*T. aestivum*, *T. dicoccoides*, *T. urartu*, and *A. tauschii*) that have undergone a strong selection pressure for purification during evolution, which may have helped to maintain the function of DUF668 (Figure 2C). This result implies a new possibility for the functional similarity of DUF668 in rice, cotton, and sweet potato.

According to the analysis results of protein characteristics, the majority of TaDUF668 members were predicted to show subcellular localization in the nucleus and chloroplasts. However, the pI and GRAVY values of most Group II members are larger than those of Group I (Table 2). This result indirectly indicates differences in function between the two groups. Based on 3D models of TaDUF668s, in general, the models within the group are similar, but the models between the groups are different (Figure 3). This result provides evidence for functional differentiation of proteins.

To better understand the reasons for differentiation, we proceeded with the analysis of gene structure. The structure of a gene determines its function [1]. The quantitative differences between exons and introns may provide evidence for evolutionary conservation and provide reference information for phylogeny [35]. The result of gene structure analysis displayed that the exons and introns of the same group were similar in the gene structure of

*TaDUF668s*, and the number and length of exons and introns of two groups were different, which is likely one of the reasons leading to the of function evolution and differentiation of *TaDUF668s* (Figure 3B). Avina-Padilla et al. [36] found that intron-deficient genes originated from prokaryotes and could be replicated in plant genomes. The intron-free *DUF668* gene has also been analyzed in *O. sativa* [13], *A. thaliana* [13], *G. hirsutum* [8], and *I. batatas* [14], suggesting that the DUF668 gene family is highly conserved across species. Moreover, the results manifested in the two groups of DUF668 may have different evolution methods. Protein-conserved motif analysis found that the number of motifs in Group I was larger than that in Group II on the whole (Figure 3C). Motifs 1, 2, 3, 4, 5, 6, 7, and 8 were found in most TaDUF668 family members, and their relative location distribution had certain rules. Meanwhile, phylogenetic analysis showed that *TaDUF668s* in the same subgroup contained more similar motifs, but the numbers of motifs in the two groups were different. It is speculated that the reason for this phenomenon may be the loss or addition of motifs in the evolutionary process of the TaDUF668 family. Motifs 6, 7, and 8 are the key components of DUF668, and their functions need to be further studied.

*Cis*-elements are among the main regulatory factors of gene expression and regulate the expression of related genes in response to growth and development processes and environmental changes [37]. *Cis*-acting elements can provide a basis for exploring gene expression patterns with different tissues or environmental stresses [38]. It has been reported that there is a significant positive correlation between upstream promoter region response genes and their *cis*-elements [39]. Nakashima et al. [40] found that there are several *cis*-elements in gramineous plants that can improve their tolerance to abiotic stress. Photoinductive elements are highly enriched in the rice DUF668 promoter and play an important role in the regulation of adaptive growth in rice and development to UV [13]. Notably, rice and wheat belong to the same Gramineae family, and both species share similar *cis*-elements in the DUF668 family. The results of *TaDUF668s cis*-element analysis showed that light-responsive elements were widely distributed in the TaDUF668 family (Figure 5). Therefore, combined with previous studies, it was speculated that these light-responsive elements played an important role in wheat growth regulation and stress response.

Expression profiling analysis can provide a favorable basis for the determination of gene function [8]. For instance, Xie et al. [41] analyzed the expression profiling of the trehalose gene family in winter wheat to reveal the molecular mechanism of winter wheat frost resistance and provide clues for its application in frost resistance breeding. The expression of plant stress response genes is one of the ways for plants to resist abiotic stress [1]. The results of expression profiling revealed diversified regulation models of *TaDUF668* genes in different wheat development stages and environment condition (Figure 6). Combined with the results of expression profiling, *TaDUF668* genes may increase their relative expression to resist environmental stress and pathogen invasion. In order to verify the hypotheses, we designed a RT-qRCR experiment according to reported methods [1].

To further ascertain the biological function of *TaDUF668s* in response to stress, six genes that respond to multiple stresses were selected for RT-qPCR. The results showed that *TaDUF668s* was involved in the response of wheat to PEG, NaCl, *F. graminearum*, and *P. striiformis* stresses (Figure 8). Zhong et al. [13] have shown that *OsDUF668* genes are involved in the process of rice stress response to PEG, NaCl, drought, and *P. cinerea* through a RT-qPCR experiment. Apart from that, they verified through RT-qPCR experiments that *OsDUF668s* were expressed in a tissue-specific manner and might have participated in plant hormone regulation. Zhao et al. [8] have found through RT-qPCR experiments that *GhDUF668s* play a crucial role in drought and *Verticillium wilt* stress. The reason for the similar results may be that the domain of protein family is conservative and thus similar in biological function. Combined with the results of expression profiling analysis and RT-qPCR analysis, we further suspect that these adverse stresses may increase the relative expression level of the corresponding gene to adapt to the change in environment and the damage from pathogens (Figure 8).

The miRNAs act in two ways: one is complementary to target miRNA and interferes with the expression of target genes through cleavage; the other is regulatory by inhibiting the translation process [42]. miRNAs are involved in plant growth, development, and stress response by regulating the transcripts of target genes [43]. In this study, miRNAs targeting *TaDUF668s* were identified, and the regulatory relationship between them was analyzed (Figure 7). There are 20 tae-miRNAs, including tae-miR5086, tae-miR1119, and tae-miR9652. Shi et al. [44] found that tae-miR1119 plays a significant part in plant response to drought stress by cutting mechanisms and regulating target genes. Through the integration of *cis*-element analysis, expression profiling, and miRNA target analysis of TaDUF668, some interesting clues were uncovered. For instance, the expression of *TaDUF668-13*, *-14*, and *-15* rapidly dropped in drought and heat stress, as their promoter regions lack drought response elements (MBSs). However, their promoter regions contain multiple "growth and development" elements, and they have high expression levels in growth and development stages. The three genes were targeted by tae-miRNA-1119, which is related to drought. Thus, it seems that, through responsible *cis*-elements, *TaDUF668-13*, *-14*, and *-15* are involved in wheat growth and development, and through suppression of the expression level by tae-miRNA-1119, they are responsive to drought stress. In summary, miRNAs and *cis*-elements form a complex regulatory mechanism in TaDUF668 expression.

## 5. Conclusions

In summary, a total of 31 *TaDUF668* genes were identified in this study, which were classified into two groups according to phylogenetic and evolutionary relationships. *TaDUF668s* are unevenly distributed on 18 chromosomes. DUF668 is highly conserved between wheat species and its three ancestral wheat species and has undergone strong selection pressure for purification during its evolution. Analysis of gene structure and protein-conserved motifs indicated that the TaDUF668 gene family in the same group had a similar structure. The analysis of *cis*-elements and expression patterns provided a theoretical basis for the response of the TaDUF668 gene family to wheat growth and development and stress. miRNA analysis elucidates the regulatory relationship between miRNA and *TaDUF668s* after transcription. The results of RT-qPCR analysis showed that the TaDUF668 family participated in the process of wheat response to stress.

**Supplementary Materials:** The following supporting information can be downloaded at: https://www.mdpi.com/article/10.3390/agronomy13082178/s1, Table S1: DUF668 protein sequences of five spcies; Figure S1: amplify and melt peak curve.

**Author Contributions:** Conceptualization, D.M., Z.F. and J.Y.; Data curation, X.Y.; Formal analysis, X.Y.; Funding acquisition, J.Y.; Investigation, X.Y., X.H., S.H., Y.Y. and Y.L.; Methodology, X.Y., X.H., S.H., Y.Y., Y.L. and J.Y.; Project administration, D.M., Z.F., S.G. and J.Y.; Resources, Y.L., D.M., Z.F., S.G. and J.Y.; Software, X.Y., X.H., Y.Y., Y.L., D.M. and Z.F.; Supervision, J.Y.; Validation, J.Y.; Visualization, X.Y., Y.Y., Y.L. and J.Y.; Writing—original draft, X.Y.; Writing—review and editing, X.H., S.H., Y.Y., Y.L., D.M., Z.F., S.G. and J.Y. All authors have read and agreed to the published version of the manuscript.

**Funding:** This work was partially supported by the Key Research and Development program of Hubei province, China (2022BBA0041) and the open Fund from Key Laboratory of Integrated Pests Management on Crops in Central China/Hubei Key Laboratory of Crop Diseases, Insect Pests and Weeds Control (2022ZTSJJ4).

**Data Availability Statement:** All datasets supporting the conclusions of this article are included within the article (and Supplementary Materials, Table S1). The genome data and sequences and expression profiles of *TaDUF668* genes used in the current study are available in the Wheat Whole Genome Database (https://wheat-urgi.versailles.inra.fr/Seq-Repository/Assemblies accessed on 15 August 2023). The datasets generated and analyzed during the current study are available from the corresponding author upon reasonable request.

**Conflicts of Interest:** The authors declare no conflict of interest. The funders had no role in the design of the study; in the collection, analyses, or interpretation of data; in the writing of the manuscript; or in the decision to publish the results.

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
