# Peer review of "Genome-Wide Identification, Characterization, and Expression Profiling of TaDUF668 Gene Family in Triticum aestivum"

_agronomy, doi:10.3390/agronomy13082178_

Round 1

Reviewer 1 Report

The authors identified TaDUF668 family members, characterized protein features, and analyzed their expression patterns under stress. They performed a wide project and the results are interesting. However, the manuscript has the following minor and major concerns before acceptance of the article for publication.

 Abstract:

It is better to present the results in the form of quantitative in the abstract not only descriptive

Keywords:

Should be in alphabetical order and different from title words.

Material and methods:

Did you perform any analysis of the variance?

More detail is needed for the Ta2291 reference gene.

The statistical analysis part is missing.

Results:

The curves related to genes such as melting curves and gene expression curves should be added to the text.

Were the gene expression data normal? What test was used to test the normality of the data? The results of tests should be added to the text.

In order to compare the difference between gene expressions, which statistical analysis was used?  Add detail to the text.

Minor editing of the English language required

Reviewer 2 Report

Dear Editor

Thank you for choosing me to review this manuscript,” Genome-wide identification, characterization, and expression profiling of TaDUF668 gene family in Triticum aestivum

The authors conducted this research to identify TaDUF668 family members, characterized protein features and analyze their expression patterns under several stresses, which aims to provide insights into TaDUF668 and lay a theoretical basis for further deciphering their biological function.

The paper is interesting; all parts are clear to readers and give sufficient information, I recommend accepting it after minor revision.

In the material and methods section, Materials and methods are well presented with sufficient information and can be replicated. the experimental design
appropriate to test the hypothesis. But could you add a separate part at the end to illustrate the statistical analysis and the used program?

      The author clearly showed the results and discussion.

The figures and tables properly show the data and are easy to interpret and understand.

The research conclusion dealt with presenting the well-obtained results from the research.

The authors cited references mostly from recent publications

Data Availability Statement: there are no additional files  with the manuscript  consider correcting it 

Regards

Round 2

Reviewer 1 Report

Most comments and suggestions have been satisfactorily addressed by the authors. In one case, as suggested in the previous review, the curves related to genes such as melting curves and gene expression curves should be added to the text. The authors claimed they placed it in the text. However, I did not find them

Editor:

Thank you for your suggestions. For your concern, we found the authors add the curves related to genes in the supplementary material.

Reviewer reply:

It is OK. There are no other concerns.

Minor editing of English language required